# Recent Advances in *Alternaria* Phytotoxins: A Review of Their Occurrence, Structure, Bioactivity, and Biosynthesis

**DOI:** 10.3390/jof8020168

**Published:** 2022-02-09

**Authors:** He Wang, Yanjing Guo, Zhi Luo, Liwen Gao, Rui Li, Yaxin Zhang, Hazem M. Kalaji, Sheng Qiang, Shiguo Chen

**Affiliations:** 1Weed Research Laboratory, College of Life Science, Nanjing Agricultural University, Nanjing 210095, China; 2018216001@njau.edu.cn (H.W.); 2021216003@stu.njau.edu.cn (Y.G.); 2020116003@stu.njau.edu.cn (Z.L.); gaoliwen0905@163.com (L.G.); 2019116003@njau.edu.cn (Y.Z.); wrl@njau.edu.cn (S.Q.); 2Agricultural and Animal Husbandry Ecology and Resource Protection Center, Ordos Agriculture and Animal Husbandry Bureau, Ordos 017010, China; lirui01@163.com; 3Department of Plant Physiology, Institute of Biology, Warsaw University of Life Sciences SGGW, 159 Nowoursynowska 159, 02-776 Warsaw, Poland; hazem@kalaji.pl; 4Institute of Technology and Life Sciences—National Research Institute, Falenty, Al. Hrabska 3, 05-090 Raszyn, Poland

**Keywords:** *Alternaria* toxins, HSTs, NHSTs, biological activities, biosynthesis

## Abstract

*Alternaria* is a ubiquitous fungal genus in many ecosystems, consisting of species and strains that can be saprophytic, endophytic, or pathogenic to plants or animals, including humans. *Alternaria* species can produce a variety of secondary metabolites (SMs), especially low molecular weight toxins. Based on the characteristics of host plant susceptibility or resistance to the toxin, *Alternaria* phytotoxins are classified into host-selective toxins (HSTs) and non-host-selective toxins (NHSTs). These *Alternaria* toxins exhibit a variety of biological activities such as phytotoxic, cytotoxic, and antimicrobial properties. Generally, HSTs are toxic to host plants and can cause severe economic losses. Some NHSTs such as alternariol, altenariol methyl-ether, and altertoxins also show high cytotoxic and mutagenic activities in the exposed human or other vertebrate species. Thus, *Alternaria* toxins are meaningful for drug and pesticide development. For example, AAL-toxin, maculosin, tentoxin, and tenuazonic acid have potential to be developed as bioherbicides due to their excellent herbicidal activity. Like altersolanol A, bostrycin, and brefeldin A, they exhibit anticancer activity, and ATX V shows high activity to inhibit the HIV-1 virus. This review focuses on the classification, chemical structure, occurrence, bioactivity, and biosynthesis of the major *Alternaria* phytotoxins, including 30 HSTs and 50 NHSTs discovered to date.

## 1. Introduction

The fungal genus *Alternaria* is a widespread and successful group growing in diverse environments worldwide, ranging from saprophytes to pathogens and even endophytes. The genus *Alternaria* was identified in the year 1816 [1]. Currently, about 300 species have been described based on phylogenetic and morphological studies, which have been further divided into 26 sections [2,3,4]. As an outstanding group of fungal pathogens, *Alternaria* species can either cause diseases in a wide range of economically important crops [1], resulting in significant economic losses, or affect human and animal health, such as through upper respiratory tract infections and asthma [4,5].

To date, over 70 toxins with different chemical structures and behaviors are known to be produced by *Alternaria* species [6]. These toxins often exhibit a variety of bioactivities, such as phytotoxic, cytotoxic, and antimicrobial properties, etc. Generally, *Alternaria* phytotoxins are divided into host-selective toxins (HSTs) and non-host-selective toxins (NHSTs) based on the susceptibility or resistance of the host. HSTs are toxic only to host plants. In contrast, NHSTs can affect many plants, regardless of whether they are a host or non-host of the pathogen producing them [7]. Most HSTs have been considered as pathogenicity factors required for fungi to invade tissues and cause disease. On the other hand, NHSTs may contribute to the development of symptoms and the proliferation of plant pathogens [8,9].

Here, we review the toxins produced by *Alternaria* spp. and summarize the classification, occurrence, mode of action, biological activity, biosynthesis, and development value of each toxin. The phytotoxins presented in the paper will be termed “toxins”, and those toxic to animals will be termed “mycotoxins”.

## 2. Host-Selective Toxins

In this section, we reviewed 30 HSTs of *Alternaria* and summarised the related pathotypes, diseases caused, chemical properties, targets in plant organelles, and biosynthetic pathways of these toxins (Table 1). Based on their chemical structures, the HSTs of *Alternaria* can be classified into seven classes: (1) epoxy-decatrienoic acid (AK-toxins, AF-toxins, and ACT-toxins); (2) sphinganine analogue (AAL-toxins); (3) pyranones (ACR-toxins); (4) cyclic peptide (AM-toxins, destruxin B, and HC-toxin); (5) tetrapeptide (AS-I toxin); (6) diketopiperazine (maculosin); and (7) ribosomal peptide (ABR-toxin). In fact, the classes (4), (5), and (6) also fall into the larger family of non-ribosomal peptides.

### 2.1. AK-Toxins, AF-Toxins, and ACT-Toxins

AK-toxins produced by the Japanese pear pathotype of *A. alternata* f. sp. *Kikuchana* were first described in Japanese pear black spot disease [9,10,11]. The same researchers identified the chemical structure, absolute configuration, and biological activity of these toxins [10]. AK-toxins are the esters of 9,10-epoxy-8-hydroxy-9-methyl-decatrienoic acid (EDA), which are the derivative of phenylalanine and hydroxyldecartienoic acid. AK-toxins consist of two types, AK-toxins I and II. Both are also mixtures of three geometric isomers, namely type-a (*2E*, *4E*, *6Z*), type-b (*2E*, *4Z*, *6E*), and type-c (*2E*, *4E*, *6E*). For each compound, the main geometry is type-b (Figure 1a) [30]. Both toxins showed toxicity only in susceptible pear cultivars, and AK-toxin I was more abundant and showed higher biological activity [31,32]. In Nijisseike, a susceptible Japanese pear cultivar, the concentration that caused venous necrosis was 5 nM of AK-toxin I or 100 nM of AK-toxin II. However, at 0.1 mM of AK-toxins I and II, there was no effect on the leaves of a resistant cultivar such as Chojuro [31].

*Alternaria* black spot disease of strawberry was first reported in 1977 and the causal pathogen was identified as *A. alternata* strawberry pathotype (*A. alternata* f. sp. *fragariae*) [12]. The pathogen produces three key molecules, AF-toxins I, II, and III (Figure 1b). AF-toxins I and II were isolated in 1979 and AF-toxin III was isolated in 1984. The chemical structures of these three toxins were first determined in 1986 [33]. The three AF-toxins have the same EDA structures, which are very similar to the AK-toxins. The conformation of the EDA parts of the AF-toxin is type-a (*2E*, *4E*, *6Z*). Of these three toxins, AF-toxin I is toxic to strawberries and pears, AF-toxin II shows toxicity to pears only, and AF-toxin III shows high toxicity to strawberries but low toxicity to pears [34].

**Figure 1 jof-08-00168-f001:**
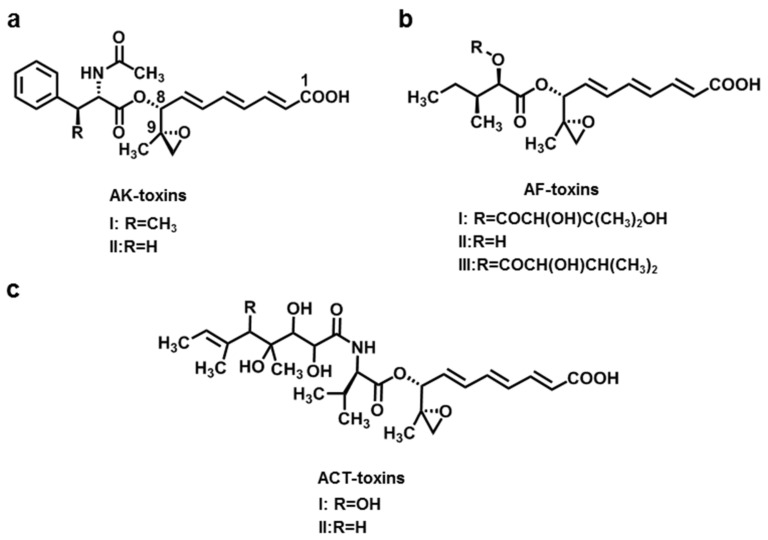
Chemical structures of AK-toxins (**a**), AF-toxins (**b**), and ACT-toxins (**c**).

*Alternaria* brown spot disease of the emperor mandarin was first reported in Australia in 1903, and the pathogen was identified as the mandarin pathotype of *A. alternata* f. sp. *citri tangerine* in 1966 [13]. This pathotype is highly toxic to mandarins, tangerines, grapefruit, and hybrids of grapefruit and tangerine, as well as mandarin and sweet orange [14]. The crucial pathogenicity depends on the action of ACT-toxins (Figure 1c). It can cause brown to black spots on young leaves, twigs, and fruits of tangerines. ACT-toxins can also be transmitted through the veins and cause more severe lesions [15]. ACT-toxins at a concentration of 2 × 10^−8^ M can cause necrotic lesions on citrus leaves with rapid electrolyte loss from the host cells. The ACT-toxins consist of three components, EDA, valine, and polyketide. ACT-toxins have two types that differ only in the R group. The conformation of the EDA component of ACT-toxins is the type B form (*2E*, *4Z*, *6E*). ACT-toxins are more abundant and toxic to citrus [35].

The target of action of AK-, AF-, and ACT-toxins is the plasma membrane of susceptible cells [34,35,36,37]. They cause a sudden, and markedly increased, K^+^ loss from the plasma membrane after a few minutes of toxin treatment, resulting in membrane invagination, vesiculation, fragmentation, and depolarization, which causes a decrease in the membrane potential gradient [37,38]. Within 1–3 h after toxin treatment, Golgi vesicles fuse with the damaged plasma membrane [37]. No damage was observed in intracellular organelles, except for the plasma membrane of host cells. Of these three toxins, AK-toxins and AF-toxins irreversibly depolarized the plasma membrane of susceptible genotypes and could directly affect the plasma membrane H^+^-ATPase [39,40,41,42]. In the case of AK-toxins, the configuration at C-8 and C-9 was critical for phytotoxicity [43].

Recently, some genes were discovered to play important biological and pathological roles in the pathotype of *A. alternata*. Two NADPH (nicotinamide adenine dinucleotide phosphate) oxidase genes (NoxA and NoxB) were identified, and NoxB was found to be essential for the aggressiveness and basal pathogenicity of *A. alternata* [44]. The gene *PEX6*, encoding a protein required for the import of matrix proteins into peroxisomes, has been characterized in *A. alternata*. It plays a role in ROS (reactive oxygen species)-induced resistance and fungal pathogenicity in the mandarin pathotype of *A. alternata* [45].

AK-toxins, AF-toxins, and ACT-toxins have a common component, EDA, in their structures [10,11,35]. In a previous study, based on the [2-^13^C]-sodium acetate feeding study of the Japanese pear pathotype of *A. alternata* and ^13^C NMR spectrum analysis, it was demonstrated that AK-toxins are biosynthesized from acetic acid via EDA [46]. In another study, ^3^H-labeled EDA was added to a growing liquid culture of the strain of the Japanese pear pathotype and was efficiently converted to AK-toxins. These results confirmed that EDA is an intermediate for toxin pathways [15].

The gene cluster involved in HST biosynthesis of *A. alternata* pathogens was first isolated from the Japanese pear pathotype, including *AKT1*, *AKT2*, *AKT3*, *AKT4*, *AKTR*, and *AKTS1* [47,48]. Recently, another gene, *AKT7*, encoding a cytochrome P450 monooxygenase was found to have the function of limiting the production of AK-toxin [7]. The biosynthetic genes of AF-toxins and ACT-toxins were identified by genomic cosmid libraries of the two pathotypes screened with the *AKT* gene probes [38]. For the biosynthetic genes of AF-toxins (*AFT*-genes), eleven *AFT*-genes and five transposon-like sequences (*TLS-S1* to *TLS-S5*) were isolated [49]. Among them, *AFT1*, *AFT3*, and *AFTR* show strong similarity to *AKT1*, *AKT3*, and *AKTR*, respectively [49]. The biosynthetic pathway of ACT-toxins was also found to be regulated by several genes, including *ACTT1*, *ACTT2*, *ACTT3*, *ACTT5*, *ACTT6*, *ACTTS2*, and *ACTTS3* [9,50,51,52], and *ACTT1* and *ACTT2* were considered to be the highly homologous genes of *AKT1* and *AKT2*, respectively, in the Japanese pear pathotype [52]. *AKT1*, *AKT2*, and *AKT3* were identified as involved in the biosynthesis of EDA, a common component of AK-, AF-, and ACT-toxins in the Japanese pear pathotype, as well as their orthologs in the strawberry and tangerine pathotypes [53]. Recently, a transcriptional regulator ACTR was identified to contribute to the biosynthesis of ACT-toxins via the mediator gene *ACTS4* in *A. alternata* [54]. These three genes were clustered on small chromosomes of less than 2.0 Mb in three pathotypic strains. They are not required for growth but confer an advantage in colonizing certain ecological niches [49,51,55,56,57].

### 2.2. AAL-Toxins

*Alternaria* stem canker disease is a serious disease of tomato (*Lycopersicon esculentum* Mill.). The disease was first described in 1975 [58]. It caused dark brown to black cankers on the stems of some tomato cultivars by a pathogenic strain, *A. alternata* f. sp. *Lycopersici* [16,17]. AAL-toxins were the main causative agent of the disease produced by the above pathogen. The first AAL-toxin was isolated in 1981 and its chemical structures, TA and TB, analogues of sphingosine and sphinganine, respectively, were determined [59,60,61]. To date, five types of AAL-toxin-related molecules, TA and TB, TC, TD, and TE, have been identified. Each of these fractions consisted of a mixture of two structural isomers (Figure 2). TA and TB showed toxicity to detached tomato leaves at 10 ng·mL^−1^. The toxicity of TD and TE is over 100 times lower than that of the form TA. The activity of TC was lower than that of TA, but higher than that of TD and TE [62]. Unlike other HSTs produced by *A. alternata*, the AAL-toxins can attack many other weeds, crops, and at least 25 species of solanaceous plants in addition to the susceptible tomato host [63,64]. On the other hand, some crops (e.g., maize, wheat, and resistant tomato varieties) are tolerant to AAL-toxins. Thus, AAL-toxins have been considered as very low-dose herbicides against a variety of broadleaf weeds such as datura, pricklesida, and black nightshade [63,65]. In addition, AAL-toxins are also toxic to cultured mammalian cells. The IC_50_ value for the most sensitive hepatoma line, H4TG, was 10 μg·mL^−1^ [66]. Such fact did limit the development of AAL-toxins as herbicides compared with some common herbicides, such as glyphosate, that are less toxic to mammals with the LD_50_ ranging from 800 to >5000 mg·kg^−1^ body weight for different animal species [67]. Recently, some AAL-toxin analogues were synthesized and one of them showed significant phytotoxicity and low mammalian toxicity, giving them potential for being developed as safe and effective natural herbicides [68,69].

When susceptible tomato leaves were treated with AAL-toxins, the accumulation of two amines, ethanolamine (EA) and phosphoethanolamine (PEA), occurred. This implies that AAL-toxins could interfere with amine metabolism [70]. When the ^14^C label of EA was fed to susceptible leaf disks treated with AAL-toxins, there was a strong inhibition of the uptake of EA into phosphatidylethanolamine (PtdEA). This phenomenon suggests possible biochemical targets of AAL-toxins, which could be enzymes involved in the phospholipid pathway [71].

Based on their chemical structure, AAL-toxins are analogous sphinganine mycotoxins (SAMTs). The SAMTs cause competitive inhibition of ceramide synthase, suppressing the conversion of sphinganine, phytosphingosine, and other free sphingoid bases into complex ceramides. The resulting accumulation of free sphingoid bases acts as a second message that activates programmed cell death (PCD) transduction pathways [72,73]. When sensitive tomato tissues were treated with AAL-toxins, sphinganine and phytosphingosine accumulated in the tissue [74]. However, this phenomenon can be avoided by ceramide supplementation, suggesting that an imbalance of ceramide is critical for triggering cell death [61,75]. Further studies have shown that both jasmonic acid (JA) and ethylene can promote AAL-toxin-induced PCD in tomato leaves by interfering with sphingolipid metabolism [76]. AAL-toxin-induced PCD is associated with ceramide signaling and cell cycle disruption. The final physiological effects of AAL-toxins are the development of necrotic lesions on fruits and leaves, the inhibition of in vitro development of calli, pollen, roots, and shoots, and the reduction of protoplast and suspension cell viability [77].

Previous studies on feeding with labeled precursors showed that glycine and the methyl group of methionine were directly incorporated into AAL-toxins. The oxygen groups in the tricarboxylic acid moieties of AAL-toxins were derived from H_2_O. The hydroxyl groups of the lipid backbone of the AAL-toxins were derived from molecular oxygen [78]. The AAL-toxin biosynthetic gene *ALT1* was identified, which encodes a type I PKS. ALT1 consists of seven domains that include *α*-ketoacyl synthase (KS), acyltransferase (AT), dehydratase (DH), methyl transferase (MT), *β*-ketoacyl reductase (KR), enoyl reductase (ER), and acyl carrier protein (ACP) [78]. Recently, a genomic BAC library of the tomato pathotype was screened using the *ALT1* probe. A 120-kb genomic region includes at least 13 genes involved in the biosynthesis of AAL-toxins. In addition to *ALT1*, the *ALT2*, *ALT3*, *ALT6*, and *ALT13* genes were also identified, encoding cytochrome P450 monooxygenase, aminotransferase, short-chain dehydrogenase/reductase, and Zn(II)2Cys6 transcription factor, respectively. *ALT* genes are located on a single small chromosome of about 1.0 Mb in the tomato pathotype strain [9].

### 2.3. ACR-Toxins

*Alternaria* brown spot disease of rough lemon was first discovered in South Africa [79]. The pathotype RLP (rough lemon) of *A. alternata* is the culprit. It can infect common citrus root species such as rough lemon (*Citrus jambhiri* Lush.) and rangpur line (*C. limonia* Osbeck) in some citrus growing areas [15,18,19]. The virulence of *A. alternata* RLP is due to the production of ACR-toxins, which may also be called ACRL-toxins [18,31,80,81]. ACR-toxins contain five compounds with different chain lengths, all of which have an *α*-pyrone group (Figure 3). The main form of ACR-toxins (ACR-toxin I, MW = 496) consists of an *α*-dihydropyrone ring in a polyalcohol with 19 carbon atoms [10,31]. ACR-toxins can cause brown necrosis on rough lemon leaves at 0.1 μg·mL^−1^, but did not affect mandarins and other non-hosts even at 1000 μg·mL^−1^ [18].

The target site of ACR-toxins is the mitochondrion, leading to mitochondrial dysfunction in rough lemon. ACR-toxins not only caused the uncoupling of mitochondrial oxidation phosphorylation, but also led to the exit of the cofactor NAD^+^ from the TCA (tricarboxylic acid) cycle [82]. The ACRS (ACR-toxin sensitivity gene), which confers sensitivity to ACR-toxins in citrus species, was identified in the mitochondrial genome of rough lemon [83]. The sensitivity was controlled by the post-transcriptional modification of the ACRS transcript.

The rough lemon pathotype strain also carried a small chromosome of 1.2–1.5 Mb, and the presence of this chromosome was associated with ACR-toxin production and rough lemon pathogenicity [51]. Several ACRT genes responsible for the biosynthesis of ACR-toxins were identified by sequence analysis of the 1.5 Mb chromosome. *ACRTS1*, *ACRTS2*, and *ACRTS3* were characterized, encoding a putative hydroxylase, a putative reducing polyketide synthase type I (PKS), and a putative cyclase, respectively. All genes were closely related to ACR-toxin production and pathogenicity [84,85]. These genes are unique to the producers of ACR-toxins of the rough lemon pathotype [86].

### 2.4. AM-Toxins

Apple cultivars such as Indo and Delicious are highly susceptible to a pathogenic strain of *A. alternata* f. sp. *mali* that can cause severe economic losses, especially in Japanese orchards [20,21]. In 1974, AM-toxins were first isolated from *A. mali*, the apple pathotype of *A. alternata* that causes apple leaf spot disease, and structural studies were conducted. AM-toxins have three distinct types (I, II, and III. Figure 4) and are produced and released by both germinating conidia and cultured mycelia of the strain. Each toxin is a four-membered cyclic depsipeptide. AM-toxin I is the most abundant among AM-toxins, causing necrosis on leaves of highly susceptible apple cultivars at concentrations of 10^−8^ M [20,21,87].

The plasma membrane and chloroplasts are two targets of AM-toxins for susceptible apple cells [88]. Similar to AK-toxins, AM-toxins can also cause plasma membrane invagination and electrolyte loss. However, the effect of AM-toxins on Japanese pear was weaker than that of AK-toxins [88]. Membrane fragments and vesicles appeared in the chloroplasts, which had emerged from grana lamellae within 3 h after toxin treatment. Chloroplast disorganization was accompanied by a decrease in chlorophyll content and inhibition of photosynthetic CO_2_ assimilation [89]. The photosynthetic activity of chloroplasts was inhibited. This phenomenon suggests that the chloroplast is a primary target of AM-toxins [38,41,90].

**Figure 4 jof-08-00168-f004:**
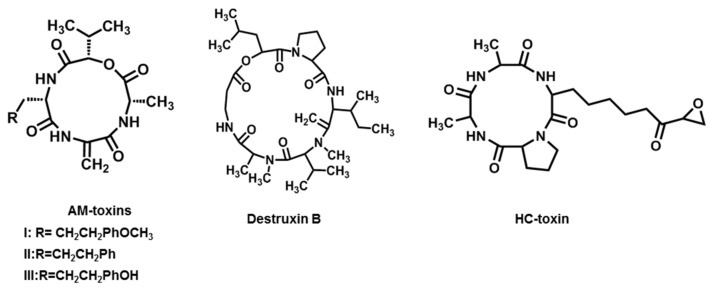
Chemical structures of AM-toxins, destruxin B, and HC-toxin.

AM-toxins belong to the cyclic peptides that are generally synthesized via non-ribosomal pathways by non-ribosomal peptide synthetases (NPRS) [91]. The AMT1, AMT2, AMT3, and AMT4 genes have been successfully isolated encoding proteins resembling enzymes involved in the secondary metabolism and modification of amino acids [92,93]. In 2007, a bacterial artificial chromosome (BAC) was isolated containing four AMT genes and other genes that are upregulated in AM-toxin-producing cultures, suggesting that genes for AM-toxin biosynthesis are clustered in the genome. It also revealed that the AMT genes are located on a conditionally dispensable (CD) chromosome of <with a size of 1.8 Mb in the strain [94].

### 2.5. Destruxin B

Black spot disease of *Brassica* spp. such as *B. campestris* and *B. napus* is caused by *A. brassicae* (Berk.) Sacc. The symptoms of the disease are lesions with grayish, brownish, or blackish centers and chlorotic margins on all above-ground parts of the plant, especially leaves, stems, and even siliques, resulting in huge economic losses in yields of about 40–60% [22,23]. The oil and protein content of the seeds is also significantly reduced, especially in *B. campestris*. Destruxin B, a cyclic peptide, is an HST that was first isolated from *A. brassicae* (Figure 4). Subsequent studies revealed that the sensitivity of *B. campestris* species to destruxin B was variable and that the order of sensitivity to destruxin B was similar to that of the pathogen. It did not cause symptoms in nine plant genera that are not hosts of *A. brassicae* [23,95]. Some researchers suggested that destruxin B may contribute to the aggressiveness of *A. brassicae* by conditioning host tissues and thereby determining host susceptibility [96].

However, in black-spot-disease resistant species (*Sinapis alba*), destruxin B could be converted into a less toxic product, hydroxydestruxin B. Essentially, hydroxydestruxin B was further biotransformed into the *β*-d-glucosyl derivative. Remarkably, it was observed that hydroxydestruxin B induced the biosynthesis of phytoalexins in black spot disease resistant species, but not in susceptible species [97].

In addition to phytotoxicity, destruxin B also exhibits a variety of biological activities. For example, significant cytotoxic effects were observed in L1210 leukemia cells and spleen lymphocytes treated with destruxin B [98]. It also showed suppressive effects on hepatitis B virus surface antigen and has been suggested as a potential candidate for the development of new anti-hepatitis agents [99,100]. Destruxin B was found to be a specific, dose-dependent, and reversible inhibitor of vacuolar ATPase, which maintains acidity in vacuolar organelles [101].

The biosynthetic pathway of destruxins, including destruxin B, was described in the fungus *Metarrhizium anisopliae*. Previously, destruxin B was thought to be biosynthesized from protodestruxin by N-methylation [102]. Feeding experiments with isotopically labeled precursors in *M. anisopliae* showed that methionine was involved in the incorporation of ^13^C into the N-methyl group of MeVal and MeAla residues. Acetates were involved in the biosynthesis of the—CH (OH)-COOH fragment of the hydroxy acid moiety, proline, and isoleucine [103].

### 2.6. HC-Toxin

When northern corn leaf spot disease was first noted in the US in 1938, it was found that *Cochliobolus carbonum* was the key pathogenic strain and could produce HC-toxin [104]. In the 1970s, Pringle and co-workers purified and partially determined the structure of HC-toxin, indicating that it was a peptide containing Ala and Pro in the ratio of 2:1 [24]. Several years later, the complete structure was established, which was cyclo (d-Pro-l-Ala-D-Ala-L-Aeo) (Figure 4), with Aeo standing for 2-amino-9,10-epoxi-8-oxodecanoic acid [25]. In 2013, HC-toxin was also found in the culture filtrates of *A. jesenskae* that was isolated from seeds of *Fumana procumbens* [26]. HC-toxin could inhibit the root growth of susceptible maize (genotype *hm1*/*hm1*) at 0.5–2 μg·mL^−1^. The concentration needed to affect resistant maize (genotype *Hm1*/-) was 100-fold higher. The epoxide group of Aeo was critical for HC-toxin toxicity, and other amino acid residues also apparently played important roles in determining the bioactivity [105]. Besides phytotoxicity, HC-toxin also showed cytostatic activity against mammalian cells. The site of action of HC-toxin was histone deacetylase (HD), an enzyme that reversibly deacetylates the core histones (H3 and H4) [106].

HC-toxin production in *C. carbonum* was controlled by a complex locus, *TOX2*, that extended over 540 kb and contained several multicopy genes. The *TOX2* locus includes *HTS1, TOXA, TOXC, TOXD, TOXE, TOXF*, and *TOXG* genes, which encoded a nonribosomal peptide synthetase, a member of the major facilitator superfamily of transporters, a fatty acid synthase beta subunit, a predicted short-chain alcohol dehydrogenase, a pathway-specific transcription factor, a putative branched chain amino acid aminotransferase, and an alanine racemase, respectively [26,107]. *A. jesenskae* had high-scoring orthologs of all known genes involved in HC-toxin biosynthesis from *C. carbonum*. Based on genomic sequencing, *AjTOX2* was considered as a major gene involved in the biosynthesis of HC-toxin in *A. jesenskae*. The genes for HC-toxin biosynthesis were duplicated in these two fungi and the encoded orthologous proteins shared 75–85% amino acid identity [26].

### 2.7. Maculosin

Spotted knapweed (*Centaurea maculosa*) is a significant threat as a weed species in North America, particularly in the northwestern United States and southwestern Canada [27,28]. Its invasion of rangelands, roadsides, and pastures has resulted in a decline in forage production of about 70% and major losses in the millions of dollars. In 1984, an infected black-leaved orchid was found in Silver Bow County (Montana, USA), and *A. alternata* was identified as the causal agent. Although seven diketopiperazines were isolated and identified (Cyclo(-l-Pro-l-Tyr-), Cyclo(-l-Pro-l-Phe-), Cyclo(-l-Pro-D-Phe-), Cyclo—Pro-Hle-), Cyclo(-Pro-Val-), Cyclo(-Pro-Leu-), and Cyclo(-Pro-Ala-)) from the liquid culture of the orchid pathogenic strain of *A. alternata*, maculosin (Cyclo(-l-Pro-l-Tyr-), Figure 5) was established as a major HST of spotted knapweed because it exhibited high toxicity to spotted knapweed at 10 μM but no toxicity to other test plants even at 1 mM [108]. Thus, it has the potential to be developed as a safe and environmentally friendly bioherbicide against knapweed.

The target site of maculosin is the chloroplasts, since within 24 h of treatment with maculosin there is a progressive decay of the chloroplasts. The core component of maculosin activity is the diketopiperazine ring, which contains proline. Subsequently, the binding component of maculosin was identified as three large molecular weight proteins, one of which was thought to be ribulose-1,5-biphosphate carboxylase (RuBPcase) [109]. Maculosin is also a potent blocker of the delayed-rectifying potassium channel in guinea pig myocytes. It can increase alkaline phosphatase expression, induce differentiation, and exert antibacterial and antioxidant effects [110,111]. To date, there is no report on the biosynthetic pathway of maculosin. A systematic, in-depth study has yet to be conducted.

### 2.8. AS-I Toxin

In 1997, two phytotoxins were isolated from the culture filtrate of *A. alternata* that are pathogenic to sunflowers [29]. The chemical structure of one toxin was deduced using chemical and physicochemical methods as tetrapeptide Ser-Val-Gly-Glu and named as AS-I toxin (Figure 5). AS-I toxin can cause chlorosis or necrosis on leaves, inhibit seed germination of sunflowers, and lead to mild toxicity on tobacco and zucchini leaves, but has no toxic effect on other plants. These phenomena suggest that AS-I toxin is an HST [29,38]. The mode of action, target, and biosynthetic pathway for AS-I toxin are still not clear, so there is a wide research scope for this HST.

### 2.9. ABR-Toxin

Most HSTs are low-molecular-weight compounds and were discovered in liquid cultures. In 2008, some researchers indicated that the spore suspensions of *A. brassicae* can cause gray leaf spot disease on *Brassica* plants. After collecting spore germination fluid (SGF) on leaves, a fraction with a high molecular weight (above 10 kDa) and toxicity to host leaves was separated by ultrafiltration. Next, a new toxin was purified from that fraction by chromatography and named ABR-toxin. Further investigation showed that ABR-toxin was a protein toxin that loses its toxicity when treated at 60 °C or with proteinase K for 15 min. The isoelectric point of ABR-toxin was about 7.0 and the molecular weight was 27.5 kDa. It contains 21 amino acid residues (Ile-Val-Gly-Gly-Val-Pro-Ala-Val-Thr-Gly-Asp-Leu-Leu-Pro-Tyr-Lys-Val-Ser-Val-Ala-Arg) with an unblocked N-terminus (Figure 5). Biological activations showed that ABR-toxin at a concentration of 0.5–1 μg·mL^−1^ could induce symptoms on *Brassica* leaves, but a concentration greater than 50 μg·mL^−1^ had no effect on non-host leaves. ABR-toxin at a concentration of 0.5–1 μg·mL^−1^ mixed with non-pathogenic spores of *A. alternata* could lead to symptoms similar to those caused by *A. brassicae* infection. The above results show that ABR-toxin not only triggered the initial colonization of host plants, but also showed a relationship with disease development that was different from that of destruxin B [23]. Currently, there are very few studies on ABR-toxin, so further detailed studies need to be conducted.

## 3. Non-Host-Selective Toxins

So far, less attention has been paid to NHSTs of *Alternaria* compared to HSTs. However, the role of NHSTs in virulence is more complex than that of HSTs. An in-depth exploration of NHSTs may reveal new and unexpected aspects for applications in many fields. Here, we detected 50 NHSTs from six families of *Alternaria*, including pyranones, quinones, tertramic acid, cyclic peptides, macrolides, and phenols (Table 2).

### 3.1. Pyranones

Pyranone is an important natural product that has attracted considerable attention due to its intriguing stereoisomeric structure and impressive bioactivity [148]. Simple pyranones and dibenzopyranones are the major groups of the pyranone family produced by *Alternaria* spp.

#### 3.1.1. Simple Pyranones

Pyranones without a benzene ring structure are defined as simple pyranones [6]. For the NHSTs of *Alternaria*, we have described here ten simple pyranones (Figure 6).

Radicinin was first found from *Stemphylium radicinum* [149] and then also isolated from *A. radicina*, including its analogue radicinol [112]. So far, many new simple pyranones NHSTs have been found in *Alternaria* spp. Radicinol and 3-epiradicinol have been isolated from other strains, such as *A. chrysanthemi*, which causes leaf spot disease in *Leucanthemum maximum* [113]. Further, 3-Epiradicinol is also found in *A. longipipes*. Deoxyradicinin was found in *A. helianthi*, an aggressive pathogen of sunflower (Figure 6a) [114].

**Figure 6 jof-08-00168-f006:**
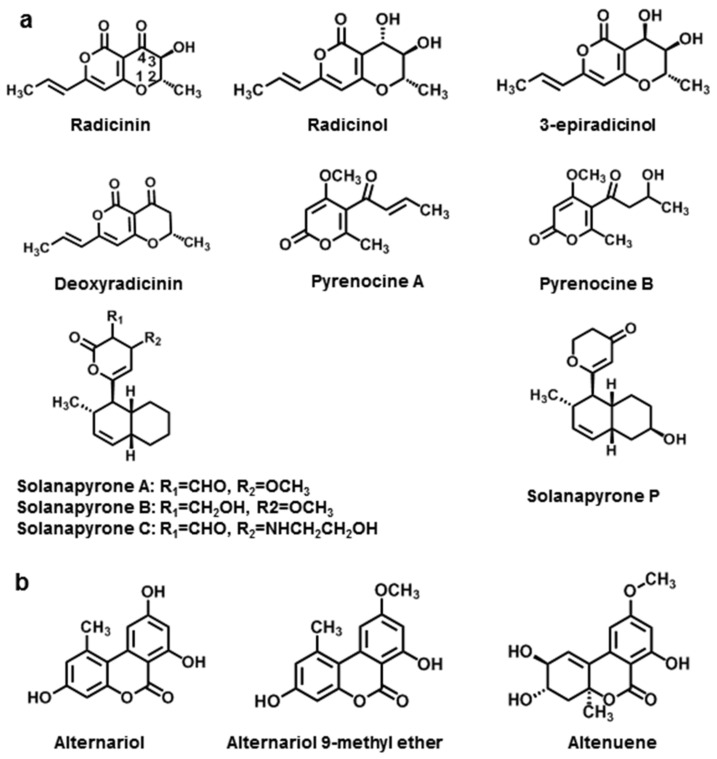
Chemical structures of *Alternaria* NHSTs belonging to simple pyranones (**a**) and dibenzopyranones (**b**) families.

Pyrenocine A and pyrenocine B were first described as products of the onion pink root fungus *Pyrenochaeta terrestris* [150]. They were then also found in the liquid medium of *A. helianthi* isolated from *Helianthus tuberosus* leaves with necrotic lesions. Pyrenocine A proved to be lethal to both isolated protoplasts and whole leaf tissue of *Helianthus* (Figure 6a) [115].

Solanapyrones A-C were isolated in 1983 from the phytopathogenic fungus *A. solani*, the causal agent of early potato blight [116]. Recently, solanapyrone P was discovered from *A. tenuissima*, an endophytic fungus in *Salvia przewalskii* (Figure 6a) [117].

Most compounds from this family showed lower phytotoxicity than radicinin [151]. Radicinin was found to be toxic to *Coix lachryma-christi* at 0.3 μg·leaf^−1^ [152]. It caused a 25% inhibition in root growth of carrot seedlings at a concentration of 10 µg·mL^−1^ [153]. In the structure of radicinin, the *α*, *β*-unsaturated carbonyl group at C-4, a free secondary hy-droxyl group at C-3, and the stereochemistry of the same carbon and the unsaturation of the propenyl side chain play key roles to exhibit activity [151]. Due to its targeting activity against the host plant and the fact that it shows no toxicity to zebrafish embryos, radicinin has the potential to be developed as a natural bioherbicide [151]. As another bioactivity, radicinin exhibits antifungal, insecticidal, and antibiotic activity against Gram-positive bacteria, including *Staphylococcus aureus* and *Clostridium* sp. [152,154].

Radicinol showed anticancer activity in various cancer cells due to modulating both tumor suppressor protein (p53) and antiapoptic protein (BCL-2), which in turn increased the expression of caspase-3 [155].

Pyrenocine A caused leaf necrosis in the leaf injury bioassay and inhibited the growth of many plants, especially greater foxglove and autumn crocus. Pyrenocine A and B induced significant electrolyte loss in the leaf tissue of bermuda grass. However, pyrenocine B showed much weaker phytotoxic activity than pyrenocine A [156]. Moreover, pyrenocine A exhibited cytotoxicity against cancer cells with an IC_50_ value of 2.6–12.9 μM [157]. Pyrenocine B inhibited the gene presentation of primary dendritic cells (DCs) in mice [158].

Solanapyrone A and B showed phytotoxicity in chickpea, resulting in stem death. Solanapyrone A was more toxic than solanapyrone B [159]. Solanapyrone A, C, and P showed antibacterial activities against various bacteria such as *Bacillus subtilis*, *B. megaterium*, *Clostridium perfringens*, *Micrococcus tetragenus*, and *Escherichia coli* with minimum inhibitory concentrations (MIC) ranging from 12.5 to 100 μg·mL^−1^ [117]. Solanapyrone A can also inhibit mammalian DNA polymerase *β* and *λ* activities in vitro, with IC_50_ values of 30 μM and 37 μM, respectively [160].

Among these simple pyranones, the biosynthetic pathway of solanapyrone A was also discovered. Feeding experiments with [1-^13^C], [1, 2-^13^C] acetates, and [S^13^CH_3_] methionine showed that solanapyrones were biosynthesized from an octaketide consisting of one acetyl-CoA, seven malonyl-CoA, and two one-carbon units from methionine [161]. Recently, a gene cluster for solanapyrone biosynthesis containing six genes, *SOL1–SOL6*, was identified for the first time in *A. solani*, suggesting that solanapyrone biosynthesis requires eight acetates and one S-adenosylmethionine (SAM) as precursors [162]. Of these genes, *SOL1* encodes a polyketide synthase that initiates the solanapyrone biosynthetic pathway, and *SOL5* encodes a Diels alderase that catalyzes both the oxidation and subsequent cyclization of the immediate precursor compound of solanapyrone A [163].

#### 3.1.2. Dibenzopyranones

The dibenzopyranone skeleton is found in many natural products and biologically active molecules. Dibenzopyranone is of great importance as an intermediate for several interesting bioactive compounds [6,164]. In this section, the three major dibenzopyranones produced by *Alternaria* are presented, namely alternariol (AOH), alternariol-9-methyl ether (AME), and altenuol (ALT) (Figure 6b).

AOH, AME, and ALT are structurally related mycotoxins produced by different *Alternaria* strains, such as *A. tenuis* [118,132]. AOH and AME were first isolated and described in 1953 [165], while ALT was discovered in 1971 [118]. Subsequent studies showed that these three compounds are present in a wide range of vegetables, fruits, mushrooms, cereals, grapes, and feeds [132,166,167]. AOH and AME are considered to be the most important *Alternaria* toxins because they are produced in relatively large amounts by most species and account for up to 20% of crude extracts of *Alternaria* isolates, while ALT accounts for only 1–3% of extracts [119,168].

AOH and AME possess broad cytotoxicity, genotoxicity, and can induce oxidative stress [169,170,171,172,173]. In vitro, AOH and AME showed cytotoxicity to Henrietta Lacks’s cervical cancer cell line HeLa cells [119]. Further studies revealed that AOH was cytotoxic to human colon carcinoma cell lines [174] and Caco-2 cells [172]. It effectively inhibited DNA relaxation and stimulated DNA cleavage activities of topoisomerase I, II*α*, and II*β* [170] and had mutagenic activity in mammalian cell lines [175]. AOH was also able to induce autophagy and senescence in murine macrophages and alter the morphology and cytokine secretion of murine and human macrophages [176,177]. In 1992, it was suggested that AOH and AME on cereals may be the most important factors for the increased incidence of human esophageal cancer in Linxian County, China [178]. Both AOH and AME appear to be highly mutagenic in the assay of *B. subtilis* and *E. coli* ND -160 [179]. Due to their widespread occurrence and high toxicity, the European Food Safety Authority (EFSA) has set the threshold of toxicological concern (TTC) for AOH and AME at 2.5 ng·kg^−1^ body weight per day [180]. ALT is most acutely toxic in female mice with a LD_50_ > 50 mg·kg^−1^ body weight, compared to AOH and AME with a LD_50_ > 400 mg·kg^−1^ body weight [181]. Recently, ALT was reported to exhibit cytotoxic activity against HCT116 cell lines with an IC_50_ value of 3.13 µM, and thus has the potential to be developed as a new antitumor drug candidate [182].

As for phytotoxic activity, AOH possessed a significant cytotoxic activity in soybean cells with an EC_50_ value of 4.69 μM. It was suggested that the phenolic hydroxyl group played a key role in the toxicity to soybean cell culture [183]. AOH inhibited root growth of *Pennisetum alopecuroides*, *Medicago sativa*, and *Amaranthus retroflexus* at 1000 μg·mL^−1^ [184]. AME inhibited the electron transport chain of spinach chloroplasts with an IC_50_ value of 29.1 μM, and inhibited the growth of *Synechococcus* by directly interacting with one or more of the electron carriers involved in the electron transport chain [185]. Although there are reports of genotoxic, estrogenic, and mutagenic effects in laboratory animals, the toxicity of AOH and AME to humans and animals is low. Thus, these compounds represent a new lead structure and have the potential to be developed as new herbicides for weed control [185].

AOH, AME, and ALT are all polyketide-derived compounds. Due to their structural similarity, the biosynthetic pathway of these compounds should be of importance. The biosynthetic pathway of AOH was first studied in detail in 1961, which suggested that AOH could be synthesized by head–tail condensations of acetate units [186]. Further studies revealed that the formation of AOH occurs by the polycondensation of malonate, which is formed by the carboxylation of acetate [187]. Later, an enzyme, alternariol-O-methytransferase from *A. alternata*, was isolated that converts AOH to AME [188]. In 2019, the gene cluster for the biosynthesis of AOH and several derivatives of *A. alternata* was found. The gene cluster contains *PSKI*, *OMTI*, *MOXI*, *SDRI*, and *DOXI*, which encode O-methyltransferase, FAD-dependent monooxygenase, short-chain dehydrogenase, putative extradiol dioxygenase, and estradiol dioxygenase, respectively. Production begins with PKSI assembling an acetyl-CoA, together with six malonyl-CoA, to form the heptaketide AOH. AOH is further converted to AME by the methyltransferase OMTI. Next, 4-hydroxy-AME is catalyzed as an intermediate by the monooxygenase MOXI, followed by the opening of the lactone ring by SDRI to form altenusin. Finally, the formation of ALT from altenusin was catalyzed by DOXI for the rotation of the C-ring and lactonization [189].

### 3.2. Quinones

Quinones are an important species that interact with biological systems to promote many beneficial agents or even induce toxicities [190]. Among *Alternaria* toxins, there are three groups of quinones, including perylenequinones, anthraquinones, and bianthraquinone derivatives that have been isolated so far. In this section, twelve perylenequinones, ten anthraquinones, and five bianthraquinones, as well as their unique bioactivities, are presented.

#### 3.2.1. Perylenequinone Derivatives

Perylenequinones are a class of aromatic polyketides characterised by a highly conjugated pentacyclic core that gives them their potent bioactivity [191]. Here, twelve perylenequinones produced by *Alternaria* are presented, including altertoxin I–VII, alterlosin I and II, alteichin, stemphyperylenol, and stemphyltoxin III.

There are many types of altertoxins (ATXs) (Figure 7). We have described seven types of ATXs from *Alternaria* spp. ATX I and ATX II were first isolated from *A. tenuis* in 1973 and ATX III was isolated from *A. alternata* in 1983 [119,120]. The correct structure of ATX I was elucidated in 1983 [120]. ATX IV was isolated from the fermentation broth of an endophytic strain of *A. tenuissima* living in the stem of *Tribulus terrestris* [121]. ATXV and VI were isolated from the fermentation broth of *A. tenuissima* QUE1Se, which inhabits the stem tissue of *Quercus emoryi* [122]. Recently, ATX VII was isolated from the endophytic fungus *Alternaria* spp. PfuH1 of patchouli (*Pogostemon cablin*). Further studies showed that all of them are perylene derivatives, which can also be produced by other *Alternaria* spp. including *A. mali* and *A. eichorniae*. Although ATXs were produced in very low amounts by only a few species, they were important *Alternaria* toxins due to their high toxicity [168]. Among them, ATX II was the most potent [175,192].

ATXs showed many activities; in particular, ATX I–III showed significant cytotoxicity, mutagenicity, and possibly carcinogenicity. In an Ames test, ATX I–III proved to be clearly mutagenic in TA98, TA100, and TA1537, with a ranking of ATX I < ATX II < ATX III [193]. ATX I and II were found to be highly toxic to the HeLa cells, with IC_50_s of 20 and 0.5 μg·mL^−1^, respectively [119]. ATX I–III were all cytotoxic to Chinese hamster V79 cells at concentrations greater than 5, 0.02, and 0.2 μg·mL^−1^, respectively [175,194]. ATX IV showed cytotoxicity to human osteosarcoma cell lines (MG-63) and human hepatocellular carcinoma cell lines (SMMC-7721), with an IC_50_ at 14.81 and 22.87 μg·mL^−1^, respectively [195]. ATX V and VI showed the ability to inhibit HIV-1 viral replication in A3.01-infected cells. ATX V showed higher activity and could completely inhibit HIV-1 virus replication at concentrations of 0.5 μM. Thus, they have the potential to be developed as potent anti-HIV drugs [122]. ATX VII showed antibacterial activities against *S. agalactiae* with MIC values of 17.3 μg·mL^−1^ [123].

As the major NHSTs of *Alternaria*, the biosynthetic pathway of ATXs was revealed. Based on the feeding experiment with ^13^C-labelled precursors, ATX I was used as an example of the biosynthetic pathway of ATXs. Five acetate molecules were found to be used for the synthesis of octalone analogues and tetralone analogues. ATXs were synthesised by the oxidative coupling of two molecules of tetralone analogues [120].

Alterlosins (ALS) include two compounds (Figure 7), ALS I and II. They were first isolated in 1989 from a host-selective strain of *A. alternata*, which is pathogenic on spotted knapweed. Both exhibited reasonable phytotoxicity, and ALS II was more potent than ALS I. ALS II was able to cause necrotic lesions on knapweed, lettuce, and Johnson grass at 10^−4^ M [124].

Alteichin (ALTCH) was isolated from *A. eichorniae* (Figure 7), a fungal pathogen of water hyacinth [120,196]. ALTCH was shown to have antifungal activity against *Valsa ceratosperma* and caused growth inhibition in lettuce seedlings [120]. Further studies revealed that ALTCH at a concentration of 0.1 mg·mL^−1^ could induce necrotic spots on the leaves of water hyacinth, tomato thistle, wheat, sunflower, and barley within 12 h. The target of ALTCH can act directly on the plant cell and cause structural changes in plant membranes [196].

Stemphyperylenol and Stemphyltoxin III could be found in the culture of *Stemphylium botryosum* and *A. alternata* (Figure 7) [124,125]. Based on the bioactivity studies, stemphyperylenol is a toxin for finger millet [197]. Stemphyltoxin III showed an in vitro antibacterial activity against *B. subtifis*, *B. cereus*, and *E. coli*, as well as phytotoxic activity (Arnone et al., 1986). SOTTX-III was also mutagenic against Ames *S. typhimurium* TA98 and TA1537 [125,198].

#### 3.2.2. Anthraquinone Derivatives

Anthraquinones (9,10-dioxoanthracenes), with the rigid planar tricyclic aromatic system anthracene, form an important class of valuable natural products [199]. There are many *Alternaria* NHSTs belonging to this family (Figure 8a).

Altersolanol A-C, E, F, and macrosporine could be isolated from *A. solani*, a pathogen of solanaceous plants. Altersolanol A and B occurred only in the culture filtrate, while the others could be isolated from either the culture filtrate or mycelia [126,127,200]. Bostrycin and 4-deoxybostrycin were isolated from the culture filtrate of *A. eichhorniae* [128]. Physcion and erythroglaucin were isolated from *A. porri* [129].

**Figure 8 jof-08-00168-f008:**
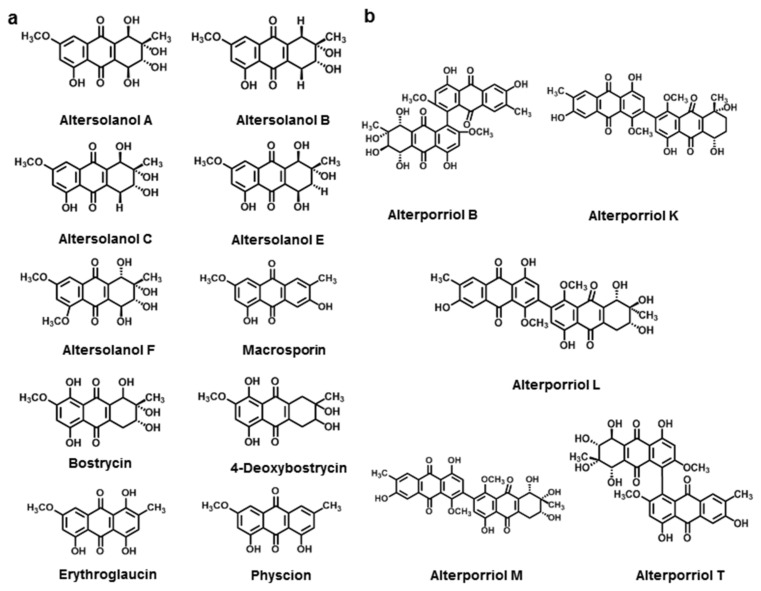
Chemical structures of *Alternaria* NHSTs belonging to anthraquinone (**a**) and bianthraquinone (**b**) families.

In bioactivity studies, altersolanols A and B showed an inhibitory effect on lettuce and stone-leek seedlings [201]. Altersolanol A-C, E, and F could act as electron transport inhibitors in the mitochondria of tobacco [202]. Altersolanol A could also cause necrosis and twisting on tomato leaves [203]. Besides phytotoxicity, altersolanols A–C and E showed antimicrobial activity against *S. aureus*, *B. subtilis*, *M. luteus*, and *Pseudomonas aeruginosa*. Altersolanols F showed obvious inhibitory activity against HCT-116 and HeLa cell lines with the IC_50_ values of 3.026 and 8.094 μM, respectively [131]. Recent studies showed that altersolanol A exhibited cytotoxicity in vitro against 34 human cancer cell lines with an IC_50_ (IC_70_) value of 0.005 μg·mL^−1^ (0.024 μg·mL^−1^). Altersolanol A was a kinase inhibitor that induced cell death by apoptosis via the cleavage of caspase-3 and -9 and a decrease in the expression of anti-apoptotic proteins [195,204].

Macrosporine exhibited antibacterial and phytotoxic activity, inhibiting *Candida albicans*, *B. subtilis*, and *S. aureus* at a dose of 200 µg·disc^−1^, and induced significant necrosis by singlet oxygenation in plants [205].

Bostrycin and 4-deoxybostrycin showed antibacterial activity against *B. subtilis*. Bostrycin was also able to inhibit the growth of *Mycobacterium tuberculosis* in vitro and inhibit the activity of effector protein tyrosine phosphatase (MptpB) secreted by Mtb. In addition, bostrycin also acted as an antitumor agent against various cancer cell lines [206,207,208,209]. Both toxins showed a phytotoxic effect on water hyacinth at a concentration of 7 and 30 μg·mL^−1^, respectively [128].

Physcion had various pharmacological properties such as anti-inflammatory, antimicrobial, and antitumor effects, including cytotoxic activity in HeLa, A549, HL-60, and SW680 cells [210]. Physcion showed no mutagenicity in an Ames assay with TA100 and TA2638 [211]. The phytotoxic activity of physcion showed that it inhibited root and hypocotyl growth less at 7.0 × 10^−4^ M in green amaranth and timothy [212].

Erythroglaucine showed a DPPH (1,1-diphenyl-2-picrylhydrazyl) radical scavenging property with an IC_50_ value of 62 µg·mL^−1^ [213].

Based on the incorporation experiment of ^13^C-labelled sodium acetate and acetate, ageolanol A, macrosporin, and other similar pigments of *A. porri* were formed by eight acetates, which were condensed in a head-to-tail process to generate a linear octaketide. Subsequent cyclization and enolization, decarboxylation, and oxidation produced the final anthraquinone analogues [214,215].

#### 3.2.3. Bianthraquinone

Many *Alternara* spp. can produce alterporriol, a member of the bianthraquinone derivatives (Figure 8b). Alterporriol B was first described in *A. porri* in 1984. To date, many alterporriols have been discovered in *Alternaria*. Alterporriol K, L, and M were obtained from the extracts of *Alternaria* sp. ZJ9-6B and showed moderate cytotoxic activity against MDA-MB-435 and MCF-7 cells with IC_50_ values ranging from 13.1 to 29.1 µM [130]. Alterporriol T was found in *Alternaria* sp. XZSBG-1 and showed an inhibition of *α*-glucosidase with an IC_50_ value of 7.2 μM [131].

Some evidence suggests that preanthraquinones serve as precursors for a number of dimers. Alterporriols are homodimers composed of two alterolanols. Alterporriol A, for example, is formed by the oxidative coupling of a macrosporin and an alterolanol A. Other alterporriols can also be biosynthesized by the same pathways as alterporriol A [214].

### 3.3. Tertramic Acids

Although they were isolated in the early 20th century, the various biological functions of tetramic acids (2,4-pyrrolidinediones) were not discovered until the 1960s [216]. Tenuazonic acid, 3-acetyl-5-isopropyltetramic acid, and 3-acetyl-5-isobutyltetramic acid are three classical analogues produced by *Alternara* [133].

Tenuazonic acid (TeA, (5S)-3-acetyl-5[(2S)-butan-2-yl]-4-hydroxy-1,5-dihydro-1H-pyrrol-2-one, Figure 9), an amide metabolite originally isolated from the culture filtrate of *A. tenuis*, is the simplest compound of the tetramic acids [217,218]. The structure and absolute configuration of TeA were elucidated after TeA was degraded by ozonolysis followed by acid hydrolysis [219]. Subsequently, TeA was also found in other species, such as *Phoma sorghina*, *Magnaporthe oryzae*, *Aspergillus* spp., and *Alternaria* spp., especially in *A. alternata*, *A. longipes*, and *A. tenuissima* [81,132,220,221,222,223,224]. Since its first isolation from cotton, TeA has been found in various vegetable, fruit, and crop plants contaminated with *Alternaria* [225,226,227].

TeA has long been reported to be toxic in animals, exhibiting antibacterial, antiviral, anticancer, and phytotoxicity effects [224,228,229,230,231]. The oral median lethal dose for male and female mice is 182 or 225 mg·kg^−1^ and 81 mg·kg^−1^ body weight, respectively [228,232]. It is also toxic to chicken embryos [233]. TeA inhibits protein biosynthesis by inhibiting the release of the polypeptide from the ribosome [234]. EFSA has evaluated the toxicity of TeA and set the threshold of toxicological concern at 1500 ng·kg^−1^ body weight per day [235]. The first study on the effect of TeA on plant cells and seedlings was published in 1974. TeA could not only cause a necrotic spot on rice leaves, but also showed a striking stunting effect on the seedling growth of rice plants, mung beans, radishes, and turnips, as well as on the growth of cells of soybean and rice plants grown in suspension [236]. In the last two decades, an increasing number of articles have reported its phytotoxicity. TeA showed an inhibitory activity against 4-hydroxyphenylpyruvate dioxygenase (HPPD) with an IC_50_ of 18 µM [237] and plant plasma membrane (PM) H^+^-ATPase [238]. TeA was also able to inhibit the elongation of seedling roots and shoots [239,240,241], and resulted in a significant increase in multi-nucleolus of *Vicia faba* root tip cells at 400 μg·mL^−1^ [242]. Qiang et al. found a crude extract named AAC-toxin containing 5% TeA produced by *A. alternata*, the natural pathogen of *Ageratina adenophora*, a common noxious weed worldwide. Further purification of the AAC-toxin and subsequent bioassays showed that TeA was primarily responsible for herbicidal activity. It exhibited broad spectrum weed activity. Thus, TeA had the potential to be used as a bioherbicide in cotton fields [224,243,244,245,246].

Detailed studies on the main mechanism of action of TeA phytotoxicity revealed that TeA is a novel inhibitor of photosystem II (PSII), disrupting electron flow beyond the primary quinone acceptor, Q_A_, by interacting with the D1 protein in the PSII reaction centers. The pyrrole ring, which contains an N-C=O group, is a core component of photosynthetic inhibitory activity [224,247,248]. TeA can induce a chloroplast-derived ROS burst that causes a range of irreversible cell damage, including chlorophyll degradation, lipid peroxidation, plasma membrane rupture, chromatin condensation, DNA cleavage, and organelle disinfection, eventually leading to rapid cell destruction and leaf necrosis in host plants [248]. TeA can also trigger the EXECUTER (EX) protein-dependent ^1^O_2_ pathway leading to cell death in *Arabidopsis* seedlings [249]. A recent study suggests that cell death triggered by TeA is an essential requirement for the pathogen *A. alternata* to successfully infect host plants. Production of ROS was critical for pathogen invasion, proliferation, and disease symptom formation during infection. TeA significantly increased the ability of the pathogen to undergo invasive hyphal growth and spread [250].

Most tetramic acids are naturally derived from hybrid PKS and nonribosomal peptide synthetases (NRPS) that come from polyketides and *α*-amino acids [224,251,252]. Thus, TeA was also expected to be a product of a PKS–NRPS hybrid enzyme [224,253]. Previous experiments with radioactive precursors showed that *A. tenuis* first used L-isoleucine and two acetate molecules to synthesize N-acetoacetyl-L-isoleucine. Subsequently, TeA was formed by the cyclization of N-acetoacetyl-L-isoleucine [133,216,224,254]. Recently, the TeA biosynthetic gene *TAS1* was discovered from *M. oryzae*. *TAS1* encodes the TeA biosynthetic enzyme TAS1, which is a NPRS–PKS hybrid protein consisting of a C (condensation)-A (adenylation)-PCP (peptidyl carrier protein)-KS (ketosynthase) domain structure [252,255]. It was found that the C-A-PCP domain of TAS1 condenses L-isoleucine and acetoacetyl-CoA to yield N-acetoacetyl-L-isoleucine, while the KS domain recognizes the N-acetoacetyl-L-isoleucine hybrid to initiate the cyclization reaction to produce TeA [252,255]. In 2020, the mechanism of cyclization to form the tetramic acid ring was illustrated by the KS domain of TAS1 in the course of TeA biosynthesis. TAS1-KS contains a conserved catalytic triad Cys179-His322-Asn376. The substrate N-acetoacetyl-L-isoleucine was transferred from the PCP domain to Cys 179 via a thioester bond. The substrate was positioned by a hydrogen bond to Ser 324, and then the methylene proton was abstracted by His-322, which triggered a nucleophilic attack on the thioester carbonyl to give TeA. Asn376 could stabilize the conformation of cis-N-acetoacetyl-L-isoleusin for the nucleophilic attack to form TeA [252,256].

Much like the above biosynthetic pathway of TeA, the addition of L-isoleucine could stimulate the production of 3-acetyl-5-isopropyltetramic acid and 3-acetyl-5-isobutyltetramic acid. In this biosynthetic pattern, it is possible and useful to obtain tetramic acids with different side chains at the 5-position by growing the organism in media fed with different L-amino acids. Gatenbeck and co-workers added ^14^C-carboxyl-labeled L-valine or L-leucine to the culture media of *A. tenuis*. From the culture extracts, the 5-isopropyl and the 5-isobutyl derivatives of the tetramic acids were prepared and purified, i.e., 3-acetyl-5-isopropyltetramic acid (3-AIPTA) and 3-acetyl-5-isobutyltetramic acid (iso-TeA, Figure 9) [133].

Based on a bioassay, 3-AIPTA showed phytotoxicity to a wide range of plants. It inhibited the root and shoot length of seedlings and eventually killed seedlings of both monocotyledonous and dicotyledonous weeds. 3-AIPTA was able to inhibit PSII electron transport rates and the growth of algal cells [257]. Further studies indicated that 3-AIPTA had the same target and lethal mechanism as TeA on weeds, but the herbicidal effect was much weaker compared to TeA [258].

3-Acetyl-5-isobutyltetramic acid, also called iso-tenuazonic acid (iso-TeA), was an isomer of TeA. Because of its similar chemical structure to TeA (Figure 9), the two toxins were thought to have similar toxicological relevance. Iso-TeA showed remarkable toxic effects on *Artemia salina*, with a mortality rate of 68.9% compared to 73.6% for TeA [259]. Iso-TeA also showed antibacterial effect on *B. megaterium* [230]. It also showed significant phytotoxicity, such as the inhibition of rice root growth with an ID_50_ (50% inhibitory dose) of 0.28 mM and marked browning of rice leaves at 10 mM [260].

### 3.4. Cyclic Peptides

Cyclic peptides exhibit remarkable biological activities due to their condensed structures [261]. In this section, we have introduced tentoxin and its competing derivatives, all of which belong to this family.

Tentoxin (TEN, Figure 10) is a secondary metabolite produced by several *Alternaria* species, including *A. alternata*, *A. citri*, *A. longipes*, *A. mali*, *A. porri*, and *A. tenuis* [134,135,136,137,138,139]. Based on the analysis of the acidic hydrolysis products and spectroscopic properties of the compound, it was found that tentoxin is a cyclic tetrapeptide containing glycine, L-leucine, N-methyl-L-alanine, and N-methy-L-dehydrphenylalany. The complete structure of tentoxin is cy-clo[*N*-methyl-l-alanyl-l-leucyl-(Z)-*α*,*β*-dehydro-*N*-methylphenylalanylglycyl] [262,263]. In addition to tentoxin, dihydrotentoxin (DHT) and isotentoxin (isoTEN) have also been isolated as metabolites from *Alternaria* species [253]. Tentoxin can be found in many products, including wheat, sorghum, fruit, and barley [264,265]. Therefore, the EFSA applied the toxicological threshold of concern (TTC) approach to TEN in its preliminary risk assessment, which was set at 1500 ng·kg^−1^ body weight per day [180,263].

As a phytotoxin, tentoxin was found to induce chlorosis in germinating seedlings of some dicotyledonous plants, but not in maize, tomatoes, and members of the Brassicaceae and Poaceae families. This was explained by the fact that sensitive species might possess a specific receptor site for tentoxin, resulting in the selective disruption of chloroplast function, reduction in the levels of chloroplast-specific lipids and proteins, and ultrastructural changes in chloroplasts [137,266,267]. Further studies indicated that tentoxin is a specific, non-competitive inhibitor of photophosphorylation and the site of action is associated with chloroplast F1-ATPase (CF1) [268,269]. Interestingly, ATP hydrolysis and synthesis were inhibited at a low dose of tentoxin, while ATPase activity was stimulated at high concentrations [270]. In 2002, the crystal structure of CF1 in complex with tentoxin showed that the binding site was located in a cleft at the *αβ*-subunit [269]. Recently, the study of the cryo-EM structure of CF1 in complex with tentoxin indicated that the cyclic ring of tentoxin with the charged or polar residues (*β*Asp83, *β*Thr82, *α*Arg297, and *α*Tyr271) and its isobutyl and phenyl moieties interact with the hydrophobic residues (*α*Ile63, *α*Leu65, and *α*Val75) between the *α*- and *β*-subunit, leading to a decrease in enzyme activity [271]. In addition, tentoxin also exhibits independent effects on plant metabolism, such as stomatal movements, ion uptake and translocation, and internal ion concentrations [272].

Previous studies indicated that tentoxin appeared on day 5 after inoculation with *A. alternata*, increased rapidly, and reached a maximum between days 9 and 12. After 14 days of inoculation, the synthesis decreased [273]. Methionine is the carbon donor in the biosynthesis of tentoxin and its precursor dihydrotentoxin [273,274]. In 1994, the tentoxin synthetase was isolated, which is a polyfunctional multienzyme with an integrated methyltransferase activity that contains active SH groups. The precursor amino acids were bound to the enzyme, then N-methylation and peptide extension occurred. Finally, dihydrotentoxin was formed by cyclization and then released to be converted into tentoxin [273]. Some researchers reported that the NRPS gene *CmNps3* was responsible for tentoxin biosynthesis in *C. miyabeanus*, and predicted that the gene *AaNps3* might be involved in tentoxin biosynthesis in *Alternaria* species [275]. Recently, two genes for tentoxin biosynthesis, a NPRS gene (*TES*) and a cytochrome P450 gene (*TES1*), were found in *A. alternata* [276]. *TES* encodes a protein of 5161 amino acids. *TES1* was closely associated with *TES* in a 5′-end-to-5′-end arrangement and was predicted to be involved in dehydrophenylalanine biosynthesis. Furthermore, a detailed analysis of TES revealed that it has a typical modular NRPS organization and consists of four modules with N-methyltransferase domains in both the second and fourth modules. The arrangement of the domains is A-T-C-A-M-T-C-A-T-C-A-M-T-C. TES assembles four precursor amino acids, Gly, Ala, Leu, and DPhe. The N-methylation of Ala and DPhe occurred in the N-methyltransferase domains, respectively. The condensation domain was located in the termination module of TES, which is responsible for the formation of intramolecular macrocyclization and final tentoxin release [276]. These findings are helpful for further studies on NRPS proteins in fungi and the mechanism of DPhe biosynthesis.

### 3.5. Macrolides

Brefeldin A (BFA) and its analogues 7-dehydrobrefeldin A (7-oxo-BFA) belong to the macrolide family, which possess antibiotic properties (Figure 10). Previously, BFA was isolated from *Penicillium* species and later found in *Alternaria* spp. such as *A. carthami* and *A. zinnia* [140,141]. 7-Oxo-BFA is another macrolide that is a potent phytotoxin of *A. carthami* [141]. BFA was particularly active. It could cause the rapid appearance of large necrotic patches and a 70% reduction in chlorophyll content when *Xanthium occidentale* leaves were treated with 10^−4^ M BFA [141]. One μg·mL^−1^ BFA was sufficient to inhibit both the germination and growth of tobacco pollen tubes and also cause the collapse of Golgi stacks [277]. Further studies showed that the Golgi stacks were the common target of BFA and 7-oxo-BFA. 7-Oxo-BFA was a more potent destroyer of the Golgi stacks than BFA [278]. As a macrolide, BFA also exhibited other important bioactivities, including antifungal, cytostatic, antimitotic, antiviral, and anticancer activities [279,280]. In most mammalian cells, 1–10 μg·mL^−1^ BFA not only inhibited secretion [281,282], but also caused profound morphological changes, including the decay of the Golgi apparatus and redistribution of Golgi enzymes into the endoplasmic reticulum [278]. BFA showed high cytotoxicity against HL-60, KB, Hela, MCF-7, and Spc-A-1 cell lines (IC_50_ = 1.0–10.0 ng·mL^−1^) [280]. There is no report on the biosynthetic pathway of BFA and 7-oxo-BFA so far.

Aldaulactone was a 10-membered benzenediol lactone molecule and was firstly purified from *A. dauci*, which was toxic to a large range of dicotyledonous plants, especially carrot. It was also indicated that aldaulactone was involved in both fungal pathogenicity and plant resistance mechanisms. In phytotoxicity, aldaulactone was toxic to carrot cells, inducing a delay in embryonic development and a decrease in cell viability [142]. The biological function and biosynthetic pathway of aldaulactone has not yet been defined.

### 3.6. Phenols

Zinniol (Figure 10), a member of the phenol family, was first isolated from *A. zinnia* (Starratt, 1968) and later detected in culture filtrates of *A. dauci* [143], *A. tagetica* [144], *A. solani*, *A. porri*, *A. carthami*, *A. macrospora*, and *A. cichorii* [145]. Zinniol showed a broad phytotoxic spectrum that could cause necrotic leaf damage [144,283,284]. Zinniol could act specifically on a certain class of plant calcium channels, but its target is not comparable to calcium channel blockers [285]. The two hydroxymethyl groups of zinniol are essential for its phytotoxic activity [143]. However, a few studies suggested that zinniol is not markedly phytotoxic to embryogenic cellular cultures of *Daucus carota* [286] and the leaves of *Tagetes erecta* [287] at physiological concentrations. In addition, Zinniol also showed cytotoxic activity in rat embryonic fibroblasts with an IC_50_ of 264 μg·mL^−1^ [284]. In general, there is still much room for research for this potential natural product.

α-Acetylorcinol is a resorcinol derivative that was first isolated from *C. lunata* in 1977 [288], and has also been reported as a secondary metabolite from various *Alternaria* spp., including *A. tenuissima*, *A. brassicicola*, and *A. dauci* [146]. It exhibited phytotoxic activity in many plants. It can induce necrosis to *Sida Spinosa*, *Chenopodium album*, *Ipomoea* sp., *Datura stramonium*, *Sorghum bicolor*, *S. halepense* [289], and *Nioctiana alata* [146]. α-Acetylorcinol also showed antifungal activity against *Trichophyton rubrum and A. fumigatus* [290].

There were many reports on the production of *p*-hydroxybenzoic acid by fungi, such as *A. tagetica* [147] *and A. dauci* [146]. It was also produced by *Epichloë bromicola* and *Diaporthe gulyae*, which were the phytopathogens of *Elymus tangutorum* and sunflower, respectively [291]. *p*-Hydroxybenzoic acid could inhibit the germination and root length of *Rumex crispus* [292]. Additionally, it also exhibited antibacterial [293], antioxidant [294], antifungal [295], antialgal [296], antimutagenic [297], and estrogenic activity [298].

## 4. Summary and Outlook

*Alternaria* is a ubiquitous genus in many ecosystems, consisting of saprophytic, pathogenic, and even endophytic species. Thus, they are a rich source of secondary metabolites. The production of various HSTs and NHSTs can be considered as a crucial reason for the survival of these fungi. In this review, we have listed only some parts of the toxins of the known *Alternaria* species. The great structural diversity, high potency, and exclusive mechanisms of action make these toxins extremely attractive for the discovery of their bioactivity. Many *Alternaria* toxins exhibit excellent herbicidal, antimicrobial, antitumor, and other bioactive properties. Some of them can be directly developed into drugs or pesticides, while others can serve as lead compounds for the discovery of new drugs or pesticides. However, several challenges must be overcome for their successful development as drug or pesticide candidates in the future. First, the biological activities and modes of action of most toxins are still unclear. Second, some active crude extracts need further purification to discover the exact active components. Third, the content of many toxins in *Alternaria* is low, so a deep exploration of their biosynthetic pathways is needed to increase the yield of the useful bioactive parts.

## Figures and Tables

**Figure 2 jof-08-00168-f002:**
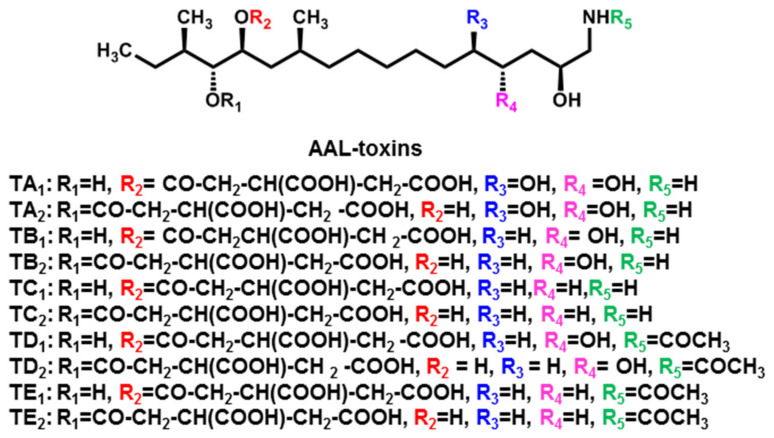
Chemical structures of AAL-toxins.

**Figure 3 jof-08-00168-f003:**
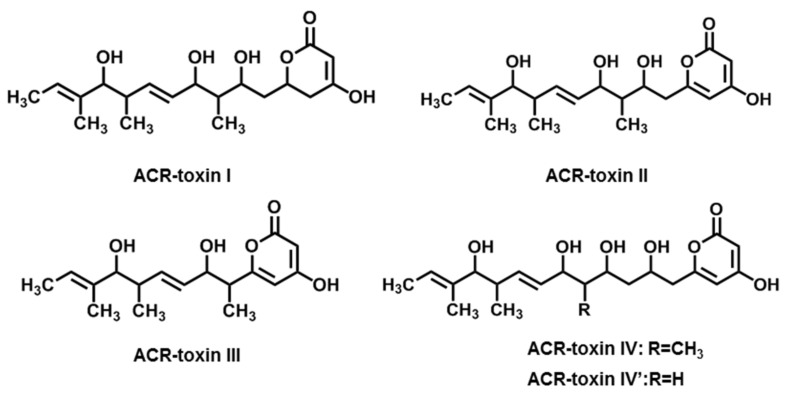
Chemical structures of ACR-toxins.

**Figure 5 jof-08-00168-f005:**
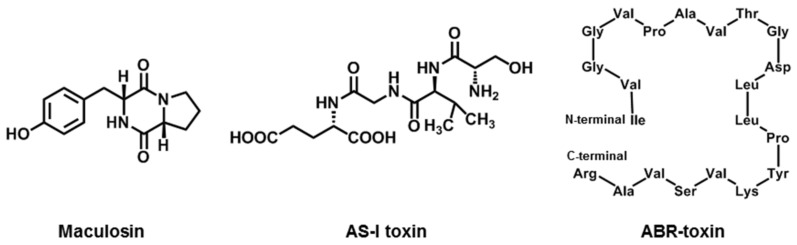
Chemical structures of maculosin, AS-I toxin, and ABR-toxin.

**Figure 7 jof-08-00168-f007:**
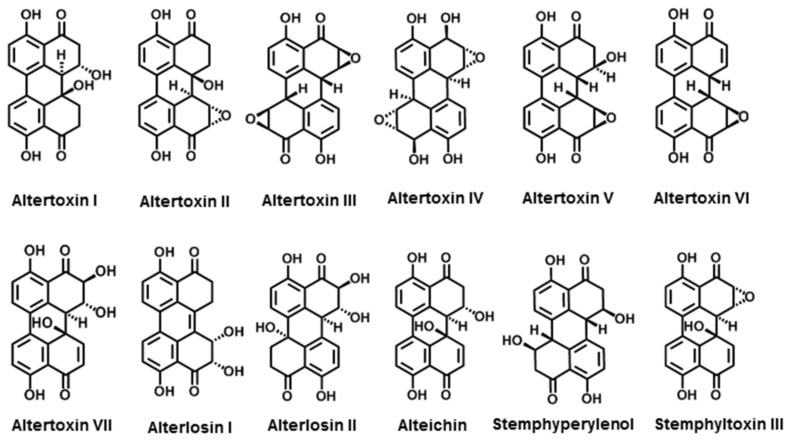
Chemical structures of *Alternaria* NHSTs belonging to perylenequinone family.

**Figure 9 jof-08-00168-f009:**
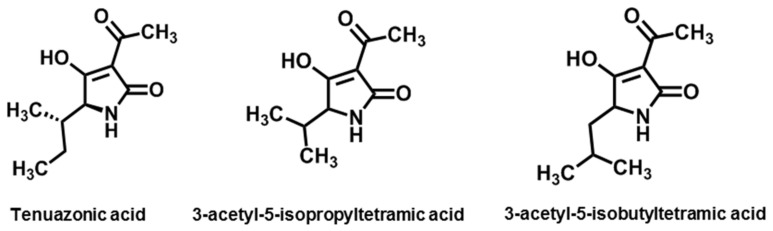
Chemical structures of *Alternaria* NHSTs belonging to the tertramic acids family.

**Figure 10 jof-08-00168-f010:**
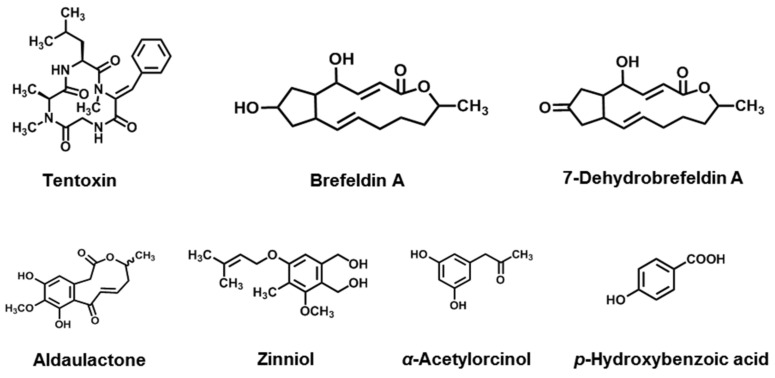
Chemical structures of *Alternaria* NHSTs belonging to the cyclic peptide (Tentoxin) macrolides (Brefeldin A, 7-dehydrobrefeldin A, and aldaulactone) and phenolics (Zinniol, *a*-acetylorcinol, and *p*-hydroxybenzoic acid) families.

**Table 1 jof-08-00168-t001:** Host-selective toxins produced by *Alternaria* species.

Toxins	*Alternaria* Species	Host Range	References
AK-toxins (AK-toxin I, II)	*A. alternata* f. sp. *kikuchana* (Japanese pear pathotype)	Japanese pear	[9,10,11]
AF-toxins (AF-toxin I, II, III)	*A. alternata* f. sp. *Fragariae* (Strawberry pathotype)	Strawberry	[12]
ACT-toxins (ACT-toxin I, II)	*A. alternata* f. sp. *citri tangerine* (Tangerine pathotype)	Tangerine	[13,14,15]
AAL-toxins (TA_1_, TA_2_, TB_1_, TB_2_, TC_1_, TC_2_, TD_1_, TD_2_, TE_1_, TE_2_)	*A. alternata* f. sp. *lycopersici* (Tomato pathotype)	Tomato	[16,17]
ACR-toxins (ACR-toxin I, II, III, IV, IV’)	*A. alternata* f. sp. *citri jambhiri* (Rough lemon pathotype)	Rough lemon	[18,19]
AM-toxins (AM-toxin I, II, III)	*A. alternata* f. sp. mali (Apple pathotype)	Apple	[20,21]
Destruxin B	*A. brassicae*	*Brassica* spp.	[22,23]
HC-toxin	*C. carbonum* and *A. jesenskae*	Maize	[24,25,26]
Maculosin	*A. alternata* (Spotted knapweed pathotype)	knapweed	[27,28]
AS-I toxin	*A. alternata* (Sunflower Pathotype)	Sunflower	[29]
ABR-toxin	*A. brassicae*	*Brassica* spp.	[23]

**Table 2 jof-08-00168-t002:** NHSTs produced by *Alternaria* species.

Family	Toxins	*Alternaria* Species	References
Pyranones	Radicinin	*A. radicina*	[112]
	Radicinol	*A. radicina*, *A. chrysanthemi*	[112,113]
	3-epiradicinol	*A. chrysanthemi, A. longipipes*	[113,114]
	Deoxyradicinin	*A. helianthi*	[114]
	Pyrenocine A	*A. helianthi*	[115]
	Pyrenocine B	*A. helianthi*	[115]
	Solanapyrones A	*A. solani*	[116]
	Solanapyrones B	*A. solani*	[116]
	Solanapyrones C	*A. solani*	[116]
	Solanapyrones P	*A. tenuissima*	[117]
	Alternariol	*A. tenuis*	[118]
	Alternariol 9-methyl ether	*A. tenuis*	[118]
	Altenuene	*A. tenuis*	[118]
Quinones	Altertoxin I	*A. tenuis*	[119]
	Altertoxin II	*A. tenuis*	[119]
	Altertoxin III	*A. alternata*	[120]
	Altertoxin IV	*A. tenuissima*	[121]
	Altertoxin V	*A. tenuissima*	[122]
	Altertoxin VI	*A. tenuissima*	[122]
	Altertoxin VII	*Alternaria* sp. PfuH1	[123]
	Alterlosins I	*A. alternata*	[124]
	Alterlosins II	*A. alternata*	[124]
	Alteichin	*A. eichorniae*	[120]
	Stemphyperylenol	*A. alternata*	[125]
	Stemphyltoxin III	*A. alternata*	[125]
	Altersolanol A	*A. solani*	[126]
	Altersolanol B	*A. solani*	[126]
	Altersolanol C	*A. solani*	[127]
	Altersolanol E	*A. solani*	[127]
	Altersolanol F	*A. solani*	[127]
	Macrosporin	*A. solani*	[126]
	Bostrycin	*A. eichhorniae*	[128]
	4-Deoxybostrycin	*A. eichhorniae*	[128]
	Physcion	*A. porri*	[129]
	Erythroglaucin	*A. porri*	[129]
	Alterporriol B	*A. porri*	[130]
	Alterporriol K	*Alternaria* sp. ZJ9-6B	[130]
	Alterporriol L	*Alternaria* sp. ZJ9-6B	[130]
	Alterporriol M	*Alternaria* sp. ZJ9-6B	[130]
	Alterporriol T	*Alternaria* sp. XZSBG-1	[131]
Tertramic acid	Tenuazonic acid	*A*. *alternata*, *A*. *longipes, A*. *tenuissima*	[132]
	3-acetyl-5-isopropyltetramic acid	*A. tenuis*	[133]
	3-acetyl-5-isobutyltetramic acid	*A. tenuis*	[133]
Cyclic peptides	Tentoxin	*A. alternata*, *A. citri*, *A. longipes*, *A. mali*, *A. porri*, *A. tenuis*	[134,135,136,137,138,139]
Macrolides	Brefeldin A	*A. carthami, A. zinnia*	[140,141]
	7-Dehydrobrefeldin A	*A. carfhami*	[141]
	Aldaulactone	*A. dauci*	[142]
Phenolics	Zinniol	*A. zinnia*, *A. dauci*, *A. tagetica*, *A. solani*, *A. porri*, *A. carthami*, *A. macrospora, A. cichorii*	[143,144,145]
	α -Acetylorcinol	*A. tenuissima, A. brassicicola, A. dauci*	[146]
	*p*-Hydroxybenzoic acid	*A. tagetica, A. dauci*	[146,147]

## Data Availability

All data presented in this study are contained in main text.

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
