# Peer review of "Recent Advances in Alternaria Phytotoxins: A Review of Their Occurrence, Structure, Bioactivity, and Biosynthesis"

_jof, 2022, doi:10.3390/jof8020168_

Round 1

Reviewer 1 Report

The topic is very interesting and within the scope of this Journal. I think is important to highlight why this review is relevant.

I did not find any ables, since these are important for a quality article, please try to add unless one.t

Abstract

From my point of view I think that it is important to talk about the toxicity. The effect of Alternaria toxins in animal, human or plant health should be described.

Furthermore, I think is important to justify this review to highlight the impact of Alteraria toxins in health and economic purposes, for example.

In this topic mycotoxins like Alternariol, Altenariol methyl-ether, tentoxin and tenuazonic acid, for axample, should be included in the abstract and should be developed in this publication…

Introduction

Lines 31 to 32: These sentences begin with Alternaria toxins, please, rephrase it.

Lines 41 and 42: “Applications of each toxin”. I don´t understand the scope of this.

Host-selective toxins

Line 60: The toxicity is not described. Please, complete it.

Non-host selctive toxins

Line 408: Please, be more concise.

Line 532-533: Anthraquinones (9,10-dioxoanthracenes) form an important class of natural products  with a variety of applications. Please, be more concise.

Author Response

The topic is very interesting and within the scope of this Journal. I think is important to highlight why this review is relevant.

RE:Thanks for your hard work and kind comments.

I did not find any tables, since these are important for a quality article, please try to add unless one.

RE:Thanks for your hard work and good suggestion. We added two tables about Alternaria HSTs and NHSTs in new version.

 Abstract

From my point of view I think that it is important to talk about the toxicity. The effect of Alternaria toxins in animal, human or plant health should be described.

Furthermore, I think is important to justify this review to highlight the impact of Alteraria toxins in health and economic purposes, for example.

In this topic mycotoxins like Alternariol, Altenariol methyl-ether, tentoxin and tenuazonic acid, for axample, should be included in the abstract and should be developed in this publication…

RE:Thanks. According to your kind advice, we revised the abstract.

Introduction

Lines 31 to 32: These sentences begin with Alternaria toxins, please, rephrase it.

RE:Thanks. We have revised these sentences.

Lines 41 and 42: “Applications of each toxin”. I don´t understand the scope of this.

RE:Thanks for your good suggestion, we have modified this sentence.

Host-selective toxins

Line 60: The toxicity is not described. Please, complete it.

RE:In fact, there is the described detail about the toxicity in Japanese pear cultivar in the end of this paragraph. As “In Nijisseike, a susceptible Japanese pear cultivar, the concentration that caused venous necrosis was 5 nM of AK-toxin I or 100 nM of AK-toxin II. However, at 0.1 mM of AK-toxins I and II there was no effect on the leaves of a resistant cultivar like Chojuro”.

Non-host selctive toxins

Line 408: Please, be more concise.

RE:Thanks for your good suggestion. We have revised this part in the revision.

Line 532-533: Anthraquinones (9,10-dioxoanthracenes) form an important class of natural products with a variety of applications. Please, be more concise.

RE:Thanks for your kind advice, we have improved this part.

Reviewer 2 Report

The authors present a knowledgeable review of the literature on the fungi Alternaria, its species, and metabolites, including 47 none host selective toxins (NHSTs) and 10 host selective toxins (HSTs). The topic of discourse is relevant and its scientific and technical aspects adequately expanded upon with relevant literature references. Overall, the manuscript is well-written.

Author Response

Thanks for your hard work and kind comments.

Reviewer 3 Report

This paper reviews the current kowlege on the phytotoxins produced by plant pathogenic fungi of the genus Alternaria, especially their prevalence, chemical structure, biological activity on plants and other organisms, and when known their biosynthesis pathways and the genes encoding the enzymes involved in it. 75 toxins are described (28 HSTs and 47 NHSTs) and seem to represent a fairly complete list of currently known toxins although a few seem to be missing. As such, this review has a high value for all researcher working with Alternaria pathogens, as it can give a complete overview on the diversity of phytotoxins produced by the Alternaria fungi.

Nevertheless, I feel like this paper is in need of various improvements. The distinction between the terms “toxin”, “mycotoxin” and “phytotoxin” have to be made more clear. Similarly, “pathogenic” and “phytopatogenic” are used as synonyms, while they are not, especially so since journal of fungi is not specialized in phytopathogenic fungi and since Alternaria fungi can be pathogenic to animals including humans. In a similar way, Alternaria alternata pathogenicity specificities are presented in a misleading way, since in several parts of the paper, it is described as specifically pathogenic to a certain plant species, or another, while the species has a huge range of host, but is also divided in many special forms, each of them specifically pathogenic to a narrow range of plant hosts, or saprophytic, or pathogenic to animals or humans. At last, the authors missed a few recently published Alternaria phytotoxins, such as the following (with refs):

HC toxin: first found in Chocliobolus carbonum Race 1, but also produced by Alternaria, see Wight et al. (2013) BMC Microbiol., 13:165 and Labuda et al. (2008) Microbiol. Res., 163(2):208-14

Aldaulactone, see Courtial et al. (2018) Front Plant Sci, 9:502

p-hydroxybenzoic acid and α-acetylorcinol, see Leyte-Lugo et al. (2020) Molecules, 25(17):4003

Here are some spelling/grammar corrections and more detailed further remarks:

Title and abstract:

Line 2: The word “toxin” should be replaced with “phytotoxin”. I suggest the following title: “Recent advances on Alternaria phytotoxins: a review of their occurrences, structure, bioactivity and biosynthesis.”

Line 12-13: “consisting of saprophytic, pathogenic, and endophytic species.” -> “consisting of species and strains that can be saprophytic, endophytic, or pathogenics to plants or animals, including humans”.

Line 14: “toxins.” -> “phytotoxins”. After that, or in the opening sentences of the introduction, the authors should add a sentence stating that the phytotoxins presented in the paper will be termed “toxins”, and those toxic to humans “mycotoxins”.

Line 17: “10 HST” -> “29 HST” Comment: 10 HST families are presented, but the total number of presented structures in fig 1-5 is 29.

  1. Introduction

I have no remarks.

  1. Host-selective toxins

Line 45: “ten HSTs” -> “29 HSTs”

Line 50: it should be highlighted that class (4), (5) and (6) fall into the larger family of non-ribosomal peptides.

Line 51: “(7) protein (ABR-toxin).” -> “(7) ribosomal peptide (ABR-toxin). Interestingly, when known, the biosynthesis genes of these toxins are found in gene clusters present in small, potentially dispensable chromosomes.”

Line 98: “con-figuration” -> “configuration”. Also, add a carriage return at the end of the sentence.

Line 102: “the gene PEX6 etc.” the link between PEX6 and toxin production/mode of action is not clear. Is PEX6 and A. alternate or a tangerine gene?

Line 124: “high-mologene” I do not understand this word.

Line 125: “coding” -> “coded”

Lines 154-7: this very small paragraph could be fused with the previous, small, paragraph. Mammal cell toxicity of AAL-toxins should be compared with data obtained from common herbicides, such as glyphosate.

Line 161: “was fed to leaf disks treated with susceptible AAL-toxins,” -> ”was fed to susceptible leaf disks treated with AAL-toxins,”

Line 163: “bi-chemical” -> “biochemical”

Line 175: “sphin-golipid” -> “sphingolipid”

Line 180: “labeling” -> “labeled”

Line 236 “the response to Japanese pear was weaker than that to AK-toxins” That is not clear. Did you mean that AM-toxin causes plasma membrane invagination to Japanese pear but this effect is weaker than with AK toxins?

Line 238: “hafter” -> h after

Line 292-3: “the orchid pathogen” -> “the orchid pathogenic strain of”

Line 297 (figure 5): N and C termini of ABR toxin should be indicated.

Line 306: Please delete the carriage return and merge this paragraph with the very short next paragraph.

Line 316: Please delete the carriage return and merge this paragraph with the very short next paragraph.

Line 326: “lost” -> “loses”

Line 336: Please delete the carriage return and merge this paragraph with the very short next paragraph.

  1. Non-host-selective toxins

Line 339: “ selctive” -> “-selective”

Line 352: “nine simple pyranones” -> “ten simple pyranones” (ten simple pyranone structures are described in fig 6a.)

Line 335: “pyranones of NHSTs” -> “pyranones NHSTs” (I believe)

Line 356: “caused” -> “causes”

Lines 357-8: “Deoxyradicinin” is not coherent with “deoxyradicinol” in fig 7. Are there two molecules, or a typo?

Line 372 : “Coix lachrymal” -> “Coix lachrima-christi

Line 386 : “bermudagrass” -> “bermuda grass”

Line 384-9: please avoid the abbreviations “Pyr A” and “Pyr B”for pyrenocine A and B, they are only used 3 times each, and not in the figure.

Lines 411-5: Please avoid the use of complete chemical names, since you did not use them for the other compounds presented here.

Line 410-1: “In this part, we have shown the three major dibenzopyranone toxins of Alternaria, including” -> “In this part, the three major 410 dibenzopyranone toxins of Alternaria are presented, namely”

Line 425: “Hela cells” -> “Henrietta Lacks’s cervical cancer cell line HeLa cells”

Line 432 “esophageal cancer in humans” -> “human esophageal cancer”

Line 432-3: an explicit year, or period of occurrence/observation/publication is needed.

Line 437: “an -> “a”

Line 438: “mg kg-1” -> “mg·kg-1

Line 457: “alter-nariol” -> “alternariol”

Line 464: “was” -> “is”

Line 466: “alternusin (ALN)” -> “alternusin”

Line 467: “ALN” -> “alternusin”

Line 472 “have demonstrated” -> “present”

Line 477: add a sentence listing altertoxins, alterlosins, alteichin, stemphyperylenone, stemphiltoxin III.

Line 479: “Alternara” -> “Alternaria” (typo, plus use italics)

Lines 481-2: “the fermentation broth of A. tenuissima, an endophytic fungal strain living in the stem of Tribulus terrestris” -> “the fermentation broth of an endophytic strain of A. tenuissima living in the stem of Tribulus terrestris

Line 499: “cervical cancer cell line Hela” -> “HeLa cells”

Line 509: “Alternara” -> “Alternaria

Line 515: “from A. alternata, a host-selective pathogen of spotted knapweed” -> “from a host-selective strain of A. alternata, pathogenic on spotted knapweed”

Line 525: “Stemphyperylenol (STPOL) and stemphyiltoxin III (STTX III)” -> “Stemphyperylenol and stemphyiltoxin III”

Line 527: “STPOL” -> “stemphyperylenol”

Line 529: “STTX III” -> “stemphyiltoxin III”

Line 540: “porri for the first time.” -> “porri.

Line 542 (Fig 8): Alterporriol B and Alterporriol T structures are not coherent with data from PubChem. For alterporriol B, there are stereochemistry discrepancies, and for alterporriol T the radicals are not the same. Please avoid such inconsistencies.

Line 546: “Alter-solanols” -> “altersolanols”

Line 547 “Alter-solanols” -> “altersolanol”

Line 546-553: I did not see data on altersolanols A-C and E phytotoxicity. Is such data available. If not, can these compounds be included in a review about phytotoxic molecules?

Line 584 and 586: Are alterporriole A and alterporriol A different or the same compound?

Line 610: “ng kg-1” -> “ng·kg-1

Line 680: “Cyclic peptide” -> “Cyclic peptides”

Line 700: “Cruciferae” -> “Brassicaceae

Line 701: “Cramineae”: Did you mean “Gramineae”? In that case, use the modern term “Poaceae”

Lines 729-30: “dehydrphenylalany” -> “dehydrophenylalanine” I guess. You should check this one in the original literature, your spelling is too incorrect for me to guess the correct term.

Lines 743-4: “A. carfhami” -> “A. carthami

Line 750: these two paragraphs can be merged.

Line 758: section title should not be the last line of a page, plus for this hierarchy, I think it should be in italics, as for 3.5 etc.

Line 764: “hy-droxymethyl” -> “hydroxymethyl”

Line 766: “torate” -> “rat” (I guess?)

Line 767: Zinniol toxicity in plants has been questioned, and also its role on the pathogenicity of A. dauci on Daucus carota (Lecomte et al. (2014) PLoS ONE 9(7): e101008) and A. tagetica on Tagetes erecta (Qui et al. (2010) J. Gen. Plant Pathol. 76: 94–101)

Line 771 “various toxins including HSTs and NHSTs” -> “various HSTs and NHSTs”

I did not check all the references in the bibliography, but I did check a few dozen, and they seem OK, except for ref 269 “Appl. Biochem. Micro+” is probably “Appl. Biochem. Microbiol.

Author Response

Dear reviewer,

We finished the review report according to your suggestion. Please find the detail in the attached file.

Best regards

Round 2

Reviewer 3 Report

I would first like to thank the authors for the extensive revision they performed on their manuscript, including the addition and careful review of new literature beyond what I suggested. In particular, the addition of tables summarizing the toxins presented in the paper is a real improvement. I think that this paper can be published as is, but I would like to indicate a few detail improvements that can still be implemented: please find them in the attached pdf document.

Author Response

I would first like to thank the authors for the extensive revision they performed on their manuscript, including the addition and careful review of new literature beyond what I suggested. In particular, the addition of tables summarizing the toxins presented in the paper is a real improvement. I think that this paper can be published as is, but I would like to indicate three detail improvements that can still be implemented:

Line 50: it should be highlighted that class (4), (5) and (6) fall into the larger family of non-ribosomal peptides.

RE: Thanks for your kind advice. We have modified this part.

RE: Apparently, you did not (lines 58-61)

RE:Thanks for your hard work and helpful comments. We are sorry for this mistake and modified this part in the new version (see Line 61 to Line 62).

Lines 154-7: this very small paragraph could be fused with the previous, small, paragraph. Mammal cell toxicity of AAL-toxins should be compared with data obtained from common herbicides, such as glyphosate.

RE:Thanks. The paragraph was fused with the previous paragraph.

The conclusion about the mammal cell toxicity of AAL-toxins is referred to the former publication (Duke, S.O.; Dayan, F.E. Clues to new herbicide mechanisms of action from natural sources. ACS Sym. Ser. 2013, 1141, 203-215) not our data. So, we thought it is not necessary to compare with the data from commercial herbicide in the review paper.

RE: I am sorry, but I beg to differ. In lines 162-5 you make the two following statement: (i) toxicity of AAL on mammal cells has been measured on H4TG hepatoma cell lines with an IC50 of 10µg/mL-1. (ii) AAL is a good candidate to produce  a  bioherbicide  that  is  “non  toxic” to  mamals.  These  two  statements  seem contradictory. In order to enlight the reader on that issue, you need to show toxicity data for glyphosate or any other common herbicide. Also, AAL analogues with lower mammal toxicity have been synthetized and tested (Abbas, H. K., Tanaka, T., & Shier, W. T. (1995). Biological activities of synthetic analogues of Alternaria alternata toxin (AAL-toxin) and fumonisin in plant and mammalian cell cultures. Phytochemistry, 40(6), 1681-1689.)

RE:Thanks for your kind advice. We have modified this part (see Line 166 to Line172).

Line 236  “the response to Japanese pear was weaker than that to AK-toxins” That is not clear. Did you mean that AM-toxin causes plasma membrane invagination to Japanese pear but this effect is weaker than with AK toxins?

RE: Thanks. We have modified this part.

RE: Thank you. Line 248  “Similar to”, you probably wanted to write  “Similarly to”.

RE: We are sorry for this mistake and modified it in the new version (see Line 252).
